# Outcomes of an intermediate respiratory care unit in the COVID-19 pandemic

Javier Carrillo Hernandez-Rubio[1]*, Maria Sanchez-Carpintero Abad[2], Andrea Yordi Leon[2], Guillermo Doblare Higuera[2], Leticia Garcia Rodriguez[2], Carmen Garcia Torrejon[3], Alejandro Mayor Cacho[4], Angel Jimenez Rodriguez[5], Mercedes Garcia-Salmones Martin[2]

1 Department of Pulmonology and Respiratory Medicine, Rey Juan Carlos University Hospital, Madrid, Spain, 2 Department of Pulmonology and Respiratory Medicine, Infanta Elena University Hospital, Madrid, Spain, 3 Department of Intensive Care Medicine, Infanta Elena University Hospital, Madrid, Spain, 4 Department of Anesthesiology, Infanta Elena University Hospital, Madrid, Spain, 5 Department of Internal Medicine, Infanta Elena University Hospital, Madrid, Spain

* jcarrillohr@gmail.com

**Data Availability Statement:** All relevant data are within the manuscript and its Supporting

## Abstract

### Background

15% of COVID-19 patients develop severe pneumonia. Non-invasive mechanical ventilation and high-flow nasal cannula can reduce the rate of endotracheal intubation in adult respiratory distress syndrome, although failure rate is high.

### Objective

To describe the rate of endotracheal intubation, the effectiveness of treatment, complications and mortality in patients with severe respiratory failure due to COVID-19.

### Methods

Prospective cohort study in a first-level hospital in Madrid. Patients with a positive polymerase chain reaction for SARS-CoV-2 and admitted to the Intermediate Respiratory Care Unit with tachypnea, use of accessory musculature or $Sp_{O2}$ <92% despite $Fi_{O2}$> 0.5 were included. Intubation rate, medical complications, and 28-day mortality were recorded. Statistical analysis through association studies, logistic and Cox regression models and survival analysis was performed.

### Results

Seventy patients were included. 37.1% required endotracheal intubation, 58.6% suffered medical complications and 24.3% died. Prone positioning was independently associated with lower need for endotracheal intubation (OR 0.05; 95% CI 0.005 to 0.54, p = 0.001). The adjusted HR for death at 28 days in the group of patients requiring endotracheal intubation was 5.4 (95% CI 1.51 to 19.5; p = 0.009).

information files. https://doi.org/10.6084/m9.
figshare.13077368.v1.

**Funding:** The authors received no specific funding
for this work.

**Competing interests:** The authors have declared
that no competing interests exist.

## Conclusions

The rate of endotracheal intubation in patients with severe respiratory failure from COVID-19 was 37.1%. Complications and mortality were lower in patients in whom endotracheal intubation could be avoided. Prone positioning could reduce the need for endotracheal intubation.

## Introduction

On March 11, 2020, the World Health Organization declared the novel COVID-19 outbreak a global pandemic [1]. More than three million cases had been reported worldwide by the end of the first week in May, of which 221,000 had been declared in Spain [2]. The spectrum of this disease caused by the SARS-CoV-2 coronavirus ranges from a common cold to a severe pneumonia defined according to American Thoracic Society criteria [3] in a not negligible 15% of patients [4]. In our setting, the rapid increase in the incidence of COVID-19 and consequent saturation of the capacity of the intensive care units (ICUs) led to a significant role for intermediate respiratory care units (IRCUs) in the management of these patients, with the principal purpose of reducing the need for endotracheal intubation (ETI) using non-invasive respiratory support.

The efficacy of high-flow nasal cannula (HFNC) and non-invasive mechanical ventilation (NIV) in adult respiratory distress syndrome (ARDS) have been previously investigated as therapies that could reduce intubation rate and mortality [5–7]. However, the use of these respiratory support therapies beyond the stablished time or severity window of the ARDS could lead to an increase in mortality [8], with failure rates in moderate or severe forms between 38% and 80% respectively [9, 10]. Hence the importance of adequate selection of patients and early access to ETI in the absence of response.

The objective of the study is to describe the ETI rate in patients with severe respiratory failure due to COVID-19 managed in an IRCU, the efficacy of the respiratory support and pharmacological treatments, and the medical complication and mortality rates.

## Material and methods

This is a prospective cohort study conducted in an 11-bed IRCU led by a team of pneumologists with support from intensive care and anesthesia specialists in the first-level Infanta Elena University Hospital, Madrid, Spain. Inclusion criteria were the following: adult patients with a positive PCR for COVID-19 and admission to the IRCU with at least one of the following: respiratory rate (RR) > 30 breaths·minute$^{-1}$, severe dyspnea, use of accessory muscles or $Sp_{O2}$ <92% despite $Fi_{O2}$ >0.5 oxygen therapy. Patients were included between March 6 and April 8, 2020 and were follow-up for a 28-day period. Patients not candidate for ETI according to the ethics committee document created and approved in March 2020 and patients transferred from the ICU to undergo weaning were excluded from the study (Fig 1).

Specific COVID-19 pharmacological treatment choice was stablished at the discretion of the prescribing specialist following the center's protocol (S1 Table).

In patients presenting a $Sp_{O2}$ <92% despite $Fi_{O2}$> 0.5 without a RR of > 30 breaths·minute$^{-1}$ or use of accessory muscles upon admission to the IRCU, treatment with HFNC (AIRVO 2, Fisher and Paykel healthcare) was started with an initial flow of 60 liters/minute, a temperature of 37.0˚C and a $Fi_{O2}$ between 0.5 and 1 with the objective of a $Sp_{O2}$> 92%.

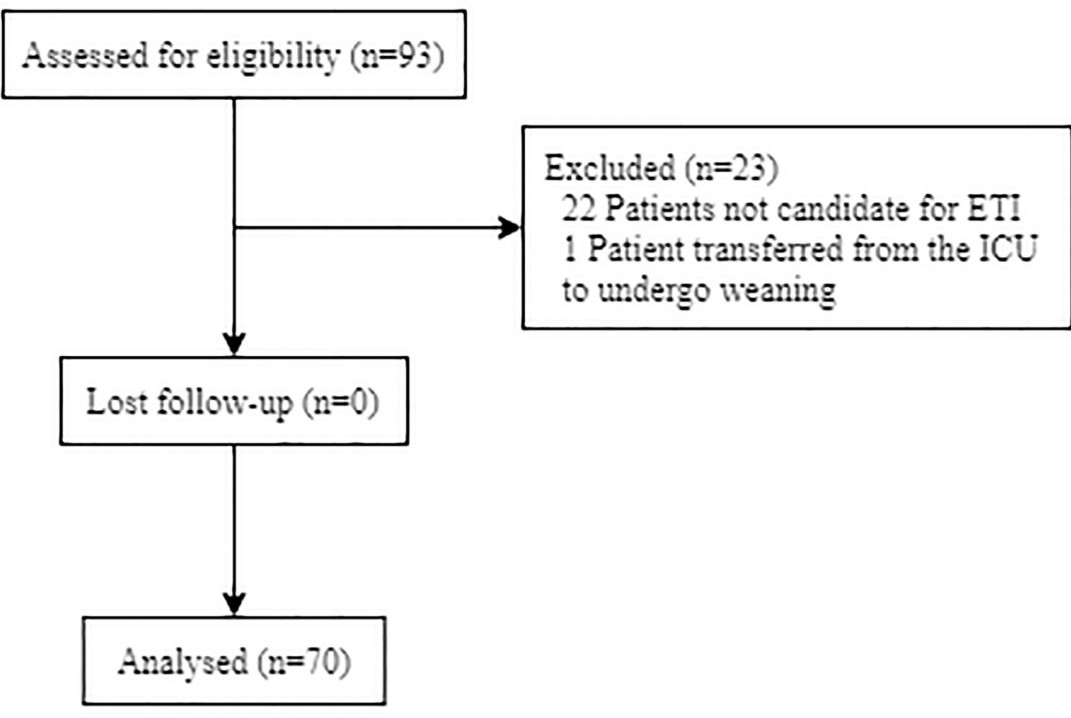

**Fig 1. Flow diagram of patient selection.** ETI: Endotracheal intubation, ICU: intensive care unit.

In patients with RR> 30 breaths·minute$^{-1}$, severe dyspnea or use of accessory muscles, support was started with IRCU or home ventilators (V60 Philips Respironics, Vivo 55 Breas, Vivo 60 Breas and Astral 150 Resmed) in CPAP or bilevel pressure mode, titrating the positive end expiratory pressure (PEEP) to achieve an $Sp_{O2}$> 92% with the lowest possible $Fi_{O2}$ and the support pressure (SP) to reduce the RR <30 breaths·minute$^{-1}$, the use of accessory muscles and the degree of dyspnea. In all cases, it was considered a priority to achieve a tidal volume of less than 6–8 ml·min$^{-1}$·kg$^{-1}$ of ideal body weight [11]. The use of Helmet was preferred for its advantages in terms of efficacy [5] and safety [12] in accordance with the recommendations of our scientific society [13]. At the discretion of the prescribing clinician, the patient was pronated in the first 12 hours of admission to the IRCU between 1 and 3 times per day for 60 minutes or as long as the patient could tolerate.

Discharge to conventional hospitalization occurred when the patient presented an $Sp_{O2}$> 92% with $Fi_{O2}$ <0.5, RR <30/min, and had no evidence of dyspnea or use of accessory muscles. ETI was performed if $Sp_{O2}$ <88%, RR> 35 breaths·minute$^{-1}$, impaired level of consciousness or hemodynamic instability, despite 1–2 hours of respiratory support with a $Fi_{O2}$> 0.5. The respiratory support algorithm is illustrated in Fig 2.

Upon admission to the IRCU, a chest radiograph was performed and a blood sample was taken for the analysis of pH, partial pressure of O2 ($Pa_{O2}$) and CO2 ($Pa_{CO2}$), and a complete biochemistry including creatinine, urea, bilirubin, liver enzymes, profile ionic, ferritin, procalcitonin, interleukin 6 (IL-6), blood count and coagulation. Age, sex, body mass index (BMI), the Charlson index and the Simplified Acute Physiology Score II (SAPS II), the choice of treatment, the highest degree of respiratory support required, the parameters of PEEP, SP and whether the patient was set in the prone position (PP) were recorded. The incidence and number of grade III, IV, and V medical complications were recorded according to the Common

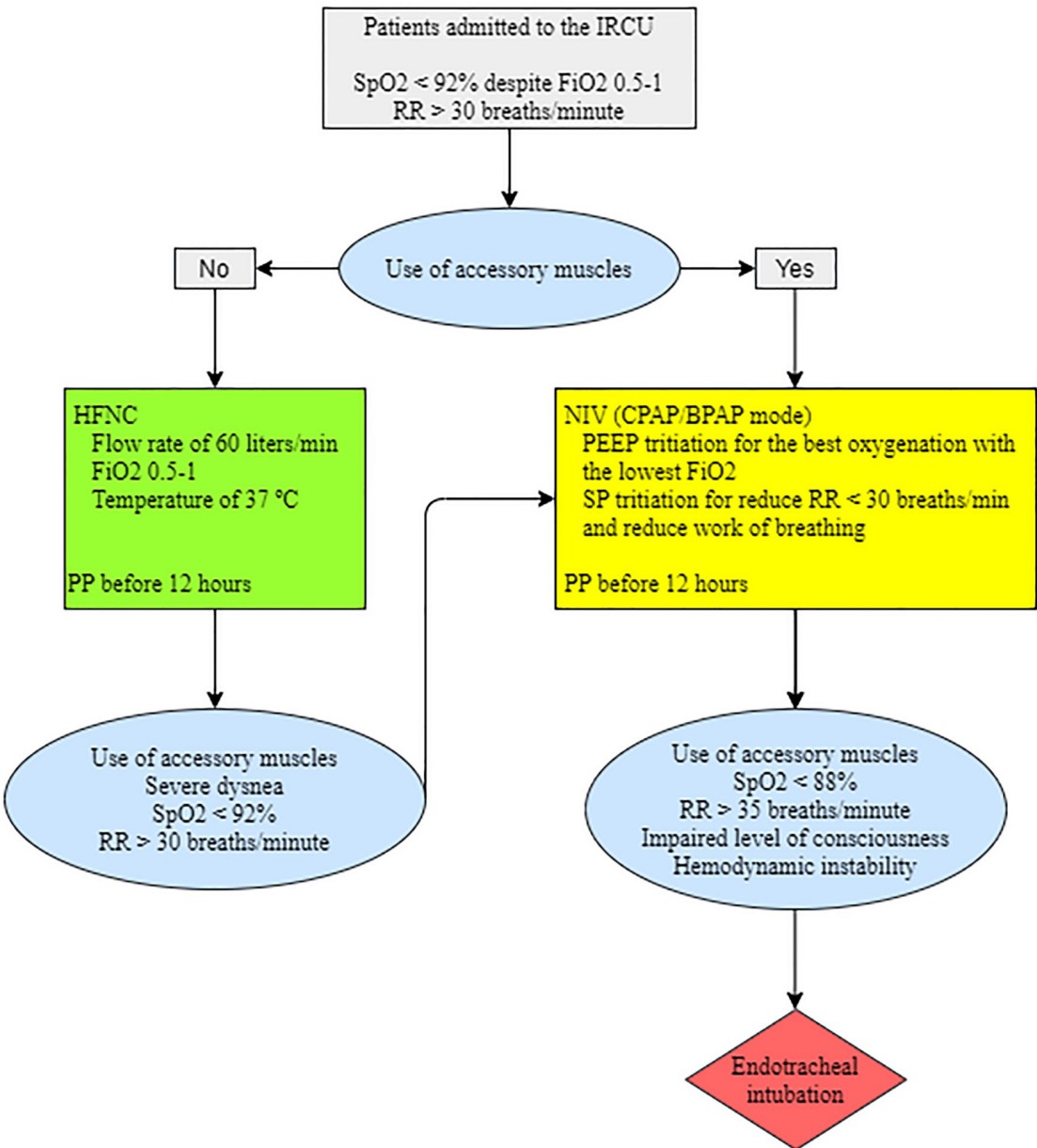

**Fig 2. Diagram of respiratory support.** IRCU: Intermediate Respiratory Care Unit, RR: respiratory rate, HFNC: high flow nasal cannula, PP: prone positioning, NIV: non-invasive ventilation, CPAP: continuous positive airway pressure, BPAP: bilevel positive airway pressure, PEEP: Positive end-expiratory pressure, SP: support pressure.

Terminology Criteria for Adverse Events (CTCAE) v5.0, as well as the intubation rate, and 28-day mortality.

The study was approved by the Research Ethics Committee of the Jiménez Díaz Foundation (code EO 081–20_HIE). The need for consent was waived by the ethics committee.

Assuming an intubation rate of 60% (9,10) in patients with moderate or severe ARDS treated with HFNC or NIV, we calculated that a sample size of at least 59 patients would be required with a two tail alpha risk of 0.05 and a statistical power of 80% in order to detect a

difference in the ETI rate of 20%, which we consider clinically relevant. A follow-up loss rate of 20% was estimated.

Descriptive statistics were obtained for all study variables. The Shapiro-Wilk test was used to verify the normality. Normally distributed quantitative variables are expressed as mean (standard deviation) or as median (interquartile range). Categorical variables are expressed as absolute and percentage value. The characteristics of the groups of intubated and non-intubated patients and deceased and non-deceased patients were compared using the Student's t-test and the Mann-Whitney U test for quantitative variables and the Chi-square test and Fisher's exact test for categorical variables. Multivariate logistic regression was performed to determine whether there were any independent predictors of ETI. Kaplan-Meier survival curves were constructed, and a log-rank analysis was conducted. We determined predictors of 28-day mortality rate using Cox proportional hazard models. For all statistical analyzes, a $p < 0.05$ was considered significant. Data were analyzed with IBM SPSS Statistics for Windows, Version 22.0. Armonk, NY: IBM Corp.

## Results

From March 6 to April 8, 2020, 93 patients were screened, of whom 70 patients were finally included in the analysis.

The median age was 60 years (range: 50.7–71.2) and 77.1% were male patients. 55.7% of patients ascertained were obese. On admission, the median $Pa_{O2}/Fi_{O2}$ was 83 mmHg (range: 55 to 142) and the mean SAPS II 34.3 ± 7.9 (SD). 55.7% of the patients required HNFC and 27.1% required NIV in CPAP or BPAP mode, and frequent Helmet use (63.2%). 37.1% of patients required ETI, 58.6% of patients suffered major medical complications, and mortality was 24.3%. The baseline characteristics of the group that required ETI and the group that did not require it are summarized in Table 1.

The levels of pH, procalcitonin, and IL-6 were significantly higher in the group that required endotracheal intubation ($p < 0.05$), while the $Pa_{CO2}$ was higher in the group of patients that did not require intubation ($p < 0.05$). No other significant differences for the analyzed variables were observed between both groups (Table 1).

**Table 1. Clinical and biochemical characteristics of the patients who required endotracheal intubation and those who did not.**

|  | Intubated (n = 26) | Not intubated (n = 44) | p-value |
|---|---|---|---|
| Age, years* | 62.5 (49.7 to 73.0) | 58.5 (51.0 to 71.0) | 0.66 |
| Male sex, No. (%) | 20 (76.9) | 31 (77.3) | 0.97 |
| Body mass index, kg·m$^{-2}$,* | 31.2 (29.8 to 34.9) | 31.3 (29.4 to 34.3) | 1 |
| Charlson Index score* | 3.0 (1.0 to 5.0) | 2.0 (1.0 to 4.0) | 0.91 |
| $Pa_{O2}/Fi_{O2}$, mmHg* | 83.5 (51.5 to 141.7) | 80.0 (64.0 to 143.0) | 0.87 |
| pH | 7.45 (0.06) | 7.40 (0.05) | 0.01 |
| $Pa_{CO2}$, mmHg* | 35.8 (6.4) | 39.9 (6.8) | 0.03 |
| SAPS II score | 33.7 (8.2) | 34.7 (7.8) | 0.62 |
| Lymphocytes cells·L$^{-1}$ | 807 (467) | 940 (249) | 0.22 |
| D dimer, µg·ml$^{-1}$,* | 309 (256 to 736) | 472 (261 to 936) | 0.83 |
| Ferritin, ng·ml$^{-1}$,* | 1752 (951 to 2939) | 1327 (708 to 1988) | 0.75 |
| Procalcitonin, ng·ml$^{-1}$,* | 0.23 (0.16 to 0.47) | 0.13 (0.06 to 0.24) | 0.03 |
| Interleukin 6, pg·ml$^{-1}$,* | 150 (54 to 1354) | 54.2 (23.4 to 88.7) | 0.02 |

SAPS II, Simple Acute Physiologic Score II.

*Data expressed as median (interquartile range).

Patients who required endotracheal intubation underwent treatment with oxygen therapy and PP in lesser proportion (p <0.05), with more frequent use of acetylcysteine, azithromycin and betaferon and less use of cyclosporine (p <0.05). No significant differences were observed in the type of respiratory support received, the ventilation parameters, the type of interface or in the rest of the pharmacological treatment used (Table 2).

Patients who required endotracheal intubation suffered a higher incidence and number of complications (p <0.05), with a higher proportion of myocardial injury, hypertension, acute kidney failure, bacteremia, septic shock, nosocomial pneumonia, bronchial obstruction, anemia, thrombopenia, and cutaneous ulcers. The mortality in patients who required intubation was significantly higher than those who did not require intubation (p <0.001) (Table 3).

A logistic regression model was designed to search for predictive variables of ETI following the backwards method. The variables age, sex, $Pa_{O_2}/Fi_{O_2}$, lymphocytes, D-dimer, procalcitonin, IL-6, pronation, the use of acetylcysteine, azithromycin, betaferon and cyclosporin were included in the model. PP was observed to be independently associated with the need for intubation (adjusted OR of 0.05, 95% CI 0.005 to 0.54, p = 0.001) (the baseline characteristics of patients who received PP and those who did not are summarized in the S3 Table) as well as IL-6 values greater than 1000 pg/ml (adjusted OR of 65.2, IC95% 3.5 to 1198, p = 0.005). In the group of patients with high levels of IL-6 (> 1000 pg·ml$^{-1}$), no significant difference in intubation rate or mortality was observed between those who received tocilizumab and those who did not.

Deceased patients were older (69 vs 57 years) and had several comorbidities (Charlson Index 6 vs 2 points) (p <0.05), lower pronation rate (17.6% vs 54.7%), greater use of betaferon (52.9 vs 24.5), lower use of cyclosporine (17.6% vs. 50.9%) and hydroxychloroquine (88.2% vs.

**Table 2. Characteristics of the support and pharmacological treatment of patients who required endotracheal intubation and those who did not.**

| | Intubated (n = 26) | Not intubated (n = 44) | p-value |
|---|---|---|---|
| Oxygen therapy, No. (%) | 1 (3.8) | 11 (25.0) | 0.02 |
| High Flow nassal cannula, No. (%) | 18 (69.2) | 21 (47.7) | 0.08 |
| CPAP, No. (%) | 2 (7.7) | 5 (11.4) | 1 |
| BPAP, No. (%) | 5 (19.2) | 7 (15.9) | 0.75 |
| Helmet, No. (%)$^{§}$ | 4 (57.1) | 8 (66.6) | 1 |
| CPAP level, cmH$_2$O* | 14.0 (13.0 to 14.0) | 15.0 (13.5 to 15.0) | 1 |
| IPAP level, cmH$_2$O | 17.2 (2.5) | 18.8 (2.6) | 0.30 |
| EPAP level, cmH$_2$O* | 10.0 (10.0 to 13.0) | 11.0 (10.0 to 12.0) | 1 |
| Prone positioning, No. (%) | 6 (23.1) | 26 (59.1) | 0.003 |
| Treatment received | | | |
| Acetylcysteine, No (%) | 21 (80.8) | 23 (52.3) | 0.01 |
| Azithromycine, No. (%) | 13 (50.0) | 9 (20.5) | 0.01 |
| Betaferón, No. (%) | 15 (57.7) | 7 (15.9) | <0.001 |
| Cyclosporine, No. (%) | 5 (19.2) | 25 (56.8) | |
| Hydroxychloroquine, No. (%) | 24 (92.3) | 44 (100) | 0.002 |
| Lopinavir/ritonavir, No. (%) | 26 (100) | 42 (95.5) | 0.13 |
| Methylprednisolone (bolus 250 mg), | 11 (42.3) | 22 (50.0) | 0.27 |
| No. (%) | | | 0.53 |
| Methylprednisolone ($\geq$ 1 mg·kg$^{-1}$·day$^{-1}$), No. (%) | 16 (61.5) | 35 (79.5) | 0.10 |
| Tocilizumab, No. (%) | 18 (69.2) | 31 (70.5) | 0.91 |

CPAP, continuous positive airway pressure; IPAP, inspiratory positive airway pressure; EPAP, Expiratory Positive Airway Pressure; BPAP, bilevel positive airway pressure.

*Data expressed as median (interquartile range).

$^{§}$From the total of patients who received non-invasive ventilation (CPAP or BPAP mode).

**Table 3. Complications and mortality of patients who required intubation and those who did not.**

| | Intubated (n = 26) | Not intubated (n = 44) | p-value |
|---|---|---|---|
| Myocardic injury, No. (%) | 8 (30.8) | 4 (9.1) | 0.046 |
| Highest cTnT value, media (DE), μg·L$^{-1}$ | 0.151 (0.06) | 0.052 (0.02) | 0.03 |
| QT prolongation, No. (%) | 3 (11.5) | 7 (15.9) | 0.73 |
| *De novo* supraventricular tachycardia, No. (%) | 3 (11.5) | 1 (2.3) | 0.15 |
| Acute pulmonary embolism, No. (%) | 1 (3.8) | 3 (6.8) | 1 |
| Hypertension, No. (%) | 7 (26.9) | 3 (6.8) | 0.03 |
| Hypotension, No. (%) | 2 (7.7) | 2 (4.5) | 0.63 |
| Acute kidney failiure, No. (%) | 9 (34.6) | 2 (4.5) | 0.002 |
| Bacteraemia, No. (%) | 4 (15.4) | 0 (0) | 0.01 |
| Septic shock, No. (%) | 5 (19.2) | 0 (0) | 0.006 |
| Hospital acquired pneumonia, No. (%) | 5 (19.2) | 0 (0) | 0.006 |
| Intravascular catheter infection, No. (%) | 3 (11.5) | 0 (0) | 0.052 |
| Fungal infection, No. (%) | 2 (7.7) | 0 (0) | 0.14 |
| Cytomegalovirus infection, No. (%) | 1 (3.8) | 0 (0) | 0.38 |
| Agitation, No. (%) | 3 (11.5) | 1 (2.3) | 0.15 |
| Seizure, No. (%) | 0 (0) | 1 (2.3) | 1 |
| Bronchial obstruction, No. (%) | 9 (34.6) | 0 (0) | <0.001 |
| Pneumothorax, No. (%) | 3 (11.5) | 0 (0) | 0.052 |
| Abnormal liver function, No. (%) | 6 (23.1) | 7 (15.9) | 0.54 |
| Hyoperkalemia, No. (%) | 1 (3.8) | 1 (2.3) | 1 |
| Hypokalemia, No. (%) | 1 (3.8) | 1 (2.3) | 1 |
| Hypernatremia, No. (%) | 1 (3.8) | 0 (0) | 0.38 |
| Hyponatremia, No. (%) | 0 (0) | 1 (2.3) | 1 |
| Hypocalcemia, No. (%) | 1 (3.8) | 1 (2.3) | 1 |
| Anemia, No. (%) | 9 (34.6) | 2 (4.5) | 0.002 |
| Thrombocytopenia, No. (%) | 4 (14.5) | 0 (0) | 0.01 |
| Gastrointestinal bleeding, No. (%) | 1 (3.8) | 1 (2.3) | 1 |
| Skin ulceration, No. (%) | 4 (15.4) | 0 (0) | 0.01 |
| Major complications, No. (%) | 21 (80.8) | 20 (45.5) | 0.001 |
| Number of major complications* | 4.5 (1.0 to 6.7) | 0.0 (0.0 to 1.7) | 0,002 |
| Mortality, No. (%) | 14 (53.8) | 3 (6.8) | <0,001 |

cTnT, cardiac troponin T.

*Data expressed as median (interquartile range).

100%) (p <0.05). The incidence of major complications in deceased patients was higher (87.5% vs. 45.5%) (p <0.05). No differences were observed in the type of support, the ventilation parameters, or other clinical and biochemical variables. The unadjusted HR for death at 28 days in the group of patients who required ETI was 9.6 (95% CI 2.7 to 33.7; p <0.001). This higher risk of death remained significant after adjusting for age, Charlson index, $Pa_{O2}/Fi_{O2}$, SAPS II, pronation, betaferon use, cyclosporine, hydroxychloroquine, and major medical complications (HR 5.4, 95% CI 1.51 to 19.5; p = 0.009) (Fig 3) (S2 Table).

## Discussion

The main finding of this study is that in 37.1% of patients with severe respiratory failure due to COVID-19 managed with non-invasive respiratory support in the IRCU ETI was needed, observing a significantly lower complication and death rates at 28 days in patients were

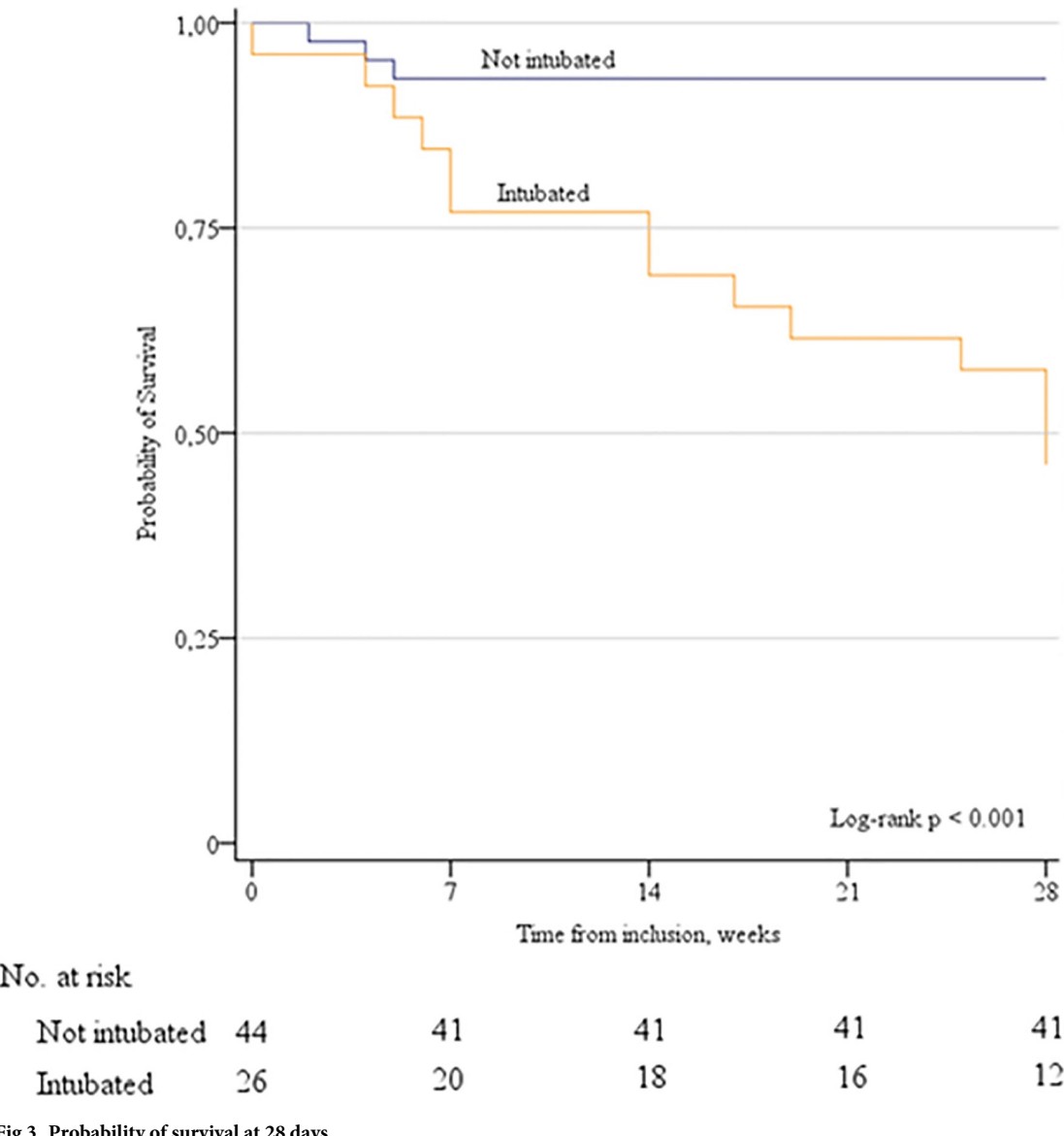

**Fig 3. Probability of survival at 28 days.**

intubation could be avoided. No therapeutic measure except PP was independently associated with intubation rate.

The efficacy of NIV in patients with de novo acute hypoxemic respiratory failure is controversial, which is why the ERS/ATS guideline only recommends its use in selected cases and by expert indication [14]. In this line, in terms of endotracheal intubation, a higher failure rate of NIV compared to oxygen therapy alone has been reported [6], attributed in part to the difficulty of avoiding ventilation-induced lung damage, [15] only achieved in 23% of cases as described by Carteaux et al. [16].

The need for ETI is understood as a severity marker that could contribute to an increased risk of complications and mortality. ETI rates in this study are similar to those previously published [17, 18] although rates may vary widely from 15% [4] to 88% [19]. However, avoiding intubation in patients with severe respiratory failure is of utmost importance given the

significant differences in terms of mortality [5, 6, 20] and associated medical complications [21]. In this study, we observed that intubation was independently associated with 28-day mortality after multivariate adjustment, as well as with a higher proportion and frequency of medical complications. The mortality rate of the patients requiring intubation was 53.8% compared to 6.8% in those in whom it could be avoided. In a meta-analysis that included 45 articles with data from 4203 patients with COVID-19, the pooled rate of mortality was 4.3% (95% CI 1.0 to 9.1) [22]. However, for patients with COVID-19 who require ICU admission, mortality rates between 16% [23] and 78% [24] have been described.

The results of this study show that PP is associated with a reduction in the need for intubation, a consistent finding in the literature [20, 25]. In a cohort study that included 20 patients with moderate or severe viral pulmonary ARDS that required support with NIV and HFNC, pronation was found to significantly improve oxygenation, as well as avoid intubation in 55% of cases [20].

The influence of the immune response to SARS-CoV-2 infection plays a key role in the severity of the disease [26]. In this line, in patients in which the local immune response is unable to resolve the infection, a dysfunctional reaction occurs that triggering a cytokine storm [27]. In these patients, high levels of plasma inflammatory mediators, including IL-6, can be observed [28, 29]. In this study, we observed that elevated plasma IL-6 levels on admission were independently associated with an increased risk of intubation. However, we did not observe a significant difference in the rate of intubation or mortality in patients with elevated IL-6 levels who received tocilizumab, an IL-6 antagonist whose efficacy is being currently evaluated in a clinical trial [30]. Although a benefit of corticosteroids in patients with COVID-19 in terms of oxygenation, symptoms, and improvement on chest CT-scan has been reported [31], we did not observe a significant difference in the rate of intubation or mortality in patients not treated with corticosteroids from those who received intravenous methylprednisolone at a dose of 1–2 mg·kg$^{-1}$·day$^{-1}$ or 250 mg boluses.

We recognize that this study has several limitations. First, the findings concerning IL-6, procalcitonin and PaO2/FiO2 ratio must be interpreted with caution because of lost data, which was assumed to be missing at random. Second, selection bias was minimized by adhering to the predefined protocol and by comparing the groups using multivariate analysis. Third, no software was used to record the tidal volume of patients treated with NIV, therefore, it could not be guaranteed in which patients a protective volume was maintained. Fourth, the duration and number of prone sessions were not recorded.

## Conclusions

The endotracheal intubation rate in patients with severe respiratory failure from COVID-19 in this study was 37.1%. The management of these patients with non-invasive respiratory support in an Intermediate Respiratory Care Units could reduce the need for intubation and consequently reduce complications and mortality. We suggest early prone positioning as part of the current respiratory therapeutic arsenal to reduce the need for endotracheal intubation.

## Supporting information

**S1 Table. Treatment protocol for patients with SARS-CoV-2 infection requiring admission to the Intermediate Respiratory Care Unit.**
(DOCX)

**S2 Table. Baseline characteristics, type of respiratory support, pharmacological treatment and complications of patients who were deceased or recovered.**
(DOCX)

**S3 Table. Clinical and biochemical characteristics of the patients who received prone positioning and those who did not.**
(DOCX)

**S1 Checklist. STROBE statement—Checklist of items that should be included in reports of cohort studies.**
(DOCX)

## Acknowledgments

Teodoro Durá Travé (Professor of Pediatrics, School of Medicine, University of Navarra, Spain), Gorka Fernández García de Eulate (Department of Neuro-Myology, Pitié-Salpêtrière Hospital, Myology Institute, Paris, France), Paula López Sánchez (Puerta de Hierro University Hospital, Madrid, España).

## Author Contributions

**Conceptualization:** Javier Carrillo Hernandez-Rubio, Maria Sanchez-Carpintero Abad, Carmen Garcia Torrejon, Mercedes Garcia-Salmones Martin.

**Data curation:** Javier Carrillo Hernandez-Rubio, Andrea Yordi Leon, Carmen Garcia Torrejon.

**Formal analysis:** Javier Carrillo Hernandez-Rubio.

**Funding acquisition:** Andrea Yordi Leon.

**Investigation:** Javier Carrillo Hernandez-Rubio, Maria Sanchez-Carpintero Abad, Guillermo Doblare Higuera, Leticia Garcia Rodriguez, Alejandro Mayor Cacho, Angel Jimenez Rodriguez, Mercedes Garcia-Salmones Martin.

**Methodology:** Andrea Yordi Leon, Guillermo Doblare Higuera, Carmen Garcia Torrejon, Alejandro Mayor Cacho, Mercedes Garcia-Salmones Martin.

**Project administration:** Angel Jimenez Rodriguez.

**Resources:** Angel Jimenez Rodriguez, Mercedes Garcia-Salmones Martin.

**Supervision:** Maria Sanchez-Carpintero Abad, Guillermo Doblare Higuera, Alejandro Mayor Cacho.

**Visualization:** Angel Jimenez Rodriguez.

**Writing – original draft:** Javier Carrillo Hernandez-Rubio, Leticia Garcia Rodriguez.

**Writing – review & editing:** Javier Carrillo Hernandez-Rubio, Leticia Garcia Rodriguez, Mercedes Garcia-Salmones Martin.

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
