## [Decision Letter · Decision Letter 0]

18 Sep 2020

PONE-D-20-22016

The role of an Intermediate Respiratory Care Unit in the COVID-19 pandemic

PLOS ONE

Dear Dr. Hernández-Rubio,

Thank you for submitting your manuscript to PLOS ONE. After careful consideration, we feel that it has merit but does not fully meet PLOS ONE’s publication criteria as it currently stands. Therefore, we invite you to submit a revised version of the manuscript that addresses the points raised during the review process.

We look forward to receiving your revised manuscript.

Kind regards,

Manjula Karpurapu

Academic Editor

PLOS ONE

Journal Requirements:

Reviewers' comments:

Reviewer's Responses to Questions

**Comments to the Author**

1. Is the manuscript technically sound, and do the data support the conclusions?

Reviewer #1: Partly

2. Has the statistical analysis been performed appropriately and rigorously? 

Reviewer #1: No

3. Have the authors made all data underlying the findings in their manuscript fully available?

Reviewer #1: Yes

4. Is the manuscript presented in an intelligible fashion and written in standard English?

Reviewer #1: Yes

5. Review Comments to the Author

Reviewer #1: Title:

“The role of an intermediate Respiratory Care Unit in the Covid-19 pandemic,”

Summary:

This cohort study, comprised of patients with severe covid infection, describes the clinical course, forms and failure rates of respiratory support administered, and outcomes of patients admitted to the intermediate various respiratory care unit (IRCU) at a single center. This study used logistic regression analyses to identify variables independently associated with endotracheal intubation. The primary findings of the study were that 37.1% of patients admitted to this IRCU progressed to invasive mechanical ventilation, prone positioning was independently associated with lower odds of invasive mechanical ventilation, and that endotracheal intubation was strongly associated with mortality.

Major Concerns:

• Generally speaking, I believe the authors need to consider alternative conclusions to the data other than the ones drawn as a result of the “inborn limitations of a cohort study.” The authors recognize this as a limitation but do not expound on this. As discussed above, the authors found that PP was independently associated with a lower odds for endotracheal intubation. The authors, while noting this association several times, also seek to infer a causative relationship. For example, the authors quote a study in the conclusion that “pronation was found to…avoid intubation in 55% of cases.” More explicitly, “we suggest early prone positioning as part of the current respiratory therapeutic arsenal to reduce the need for endotracheal intubation.” Because this was a cohort study, not all patients who received prone positioning prior to endotracheal intubation likely had the same prognosis. There was, almost absolutely, prognostic imbalance between those that could receive PP and those that could not. I suspect it is more likely that patients who could undergo prone positioning had a more favorable prognosis than those who couldn’t. A demographic table of patients who received PP and those who did not would help speak to concerns about this possible prognostic imbalance. It is not possible to conclude that PP led to reduced endotracheal intubation.

• A second concern is the often stated conclusion that stemmed from the finding that patients who underwent endotracheal intubation had a significantly worse mortality than those who didn’t. The authors state “avoiding intubation in patients with severe respiratory failure is of upmost importance given the significant differences in terms of mortality…and associated medical complications.” The authors, earlier, list bacteremia, septic shock, AKI, and MI as complications of endotracheal intubation. An alternative interpretation, which is more plausible, is that endotracheal intubation did not CAUSE these differences. Instead, patients who underwent endotracheal intubation were sicker patients with a worse prognosis. Endotracheal intubation, in other words, was a marker for severity of illness. It is not surprising that these patients then did worse. Could endotracheal intubation has contributed? Maybe. There are, absolutely, risks associated with endotracheal intubation, especially if the endotracheal tube is misplaced and intubation is delayed and face mask ventilation ineffective, for example. Attributing shock, AKI, and MI to endotracheal intubation, however, distorts the illness complexity of these patients.

• Another concern is related to the difference between what the article suggests to discuss and then what the article discusses, which also speaks to the novelty of the study. I was intrigued by the idea of the IRCU and interested in knowing if this could improve outcomes of patients. The title suggests that the focus of the article will be on the impact of the IRCU in these patients. However, the article later says that the number of patients who underwent endotracheal intubation and who died were consistent with other reports in the literature. This begs the question then if the IRCU had a positive impact or not. A conclusion was that “management of these patients in an IRCU could reduce the need for intubation and consequently reduce complications and mortality.” There is no data to support this. If true, this would be novel and reportable.

• The manuscript would be improved if the covid patients admitted to the IRCU were compared with the outcome of covid patients admitted at the same time to the ICU. Presumably, but not necessarily the ICU patients had more co-morbidities and greater severity of disease. However, if the capacity of the ICU was overwhelmed, the severity of patients admitted to the IRCU may have been closer to the severity of the concomitant ICU.

Minor Concerns:

• The authors do not explain how missing data is accounted for in the analysis. Were these patients excluded? Was data assumed to be missing at random. There is missing data for 13 patients for IL6, which is ~ 19% of the cohort. The authors then found a significant association with IL6 levels. Additional missing data includes procalcitonin (11), another significant association, and Pa02/FiO2 (15 patients), which wasn’t significant.

• Why were 22 patients not candidates for endotracheal intubation? DNI orders? Further explanation of their exclusion could be helpful.

6. PLOS authors have the option to publish the peer review history of their article (what does this mean?). If published, this will include your full peer review and any attached files.

Reviewer #1: No

---

## [Author Response · Author response to Decision Letter 0]

30 Oct 2020

1. The authors need to consider alternative conclusions to the data other than the ones drawn as a result of the “inborn limitations of a cohort study.” The authors recognize this as a limitation but do not expound on this.

Page 18, 2nd paragraph, row 13.

A comment has been added about the limitations of our study and the methods used to minimize them. First, the findings concerning IL-6, procalcitonin and PaO2/FiO2 ratio must be interpreted with caution because of lost data, which was assumed to be missing at random. Second, selection bias was minimized by adhering to the predefined protocol and by comparing the groups using multivariate analysis.

2. I suspect it is more likely that patients who could undergo prone positioning had a more favorable prognosis than those who couldn’t. A demographic table of patients who received PP and those who did not would help speak to concerns about this possible prognostic imbalance. It is not possible to conclude that PP led to reduced endotracheal intubation.

Page 15, first paragraph, row 6.

A demographic table of patients who received PP and those who did not has been added in the supplementary material. Given the absence of statistically significant differences in the prognosis of the two groups of patients and the results of the multivariate regression model, we believe that PP indeed possibly led to lower need for endotracheal intubation in our cohort of patients.

The baseline characteristics of patients who received PP and those who did not are summarized in the supplementary material (Supplementary table S3).

3. Endotracheal intubation, in other words, was a marker for severity of illness. It is not surprising that these patients then did worse. Could endotracheal intubation has contributed? Maybe. There are, absolutely, risks associated with endotracheal intubation, especially if the endotracheal tube is misplaced and intubation is delayed and face mask ventilation ineffective, for example. Attributing shock, AKI, and MI to endotracheal intubation, however, distorts the illness complexity of these patients. 

Page 17, 2nd paragraph, row 9 to 10.

We thank the reviewer for his comment and apologize if we have been unintentionally misleading. We have made the suggested changes, underlining the presence of only statistical association in endotracheal intubation and worst outcomes and making clear that this does not imply causality of intubation in terms of complications and mortality.

The need for endotracheal intubation is understood as a marker of severity, given that it could contribute to an increased risk of complications and mortality.

4 The title suggests that the focus of the article will be on the impact of the IRCU in these patients. However, the article later says that the number of patients who underwent endotracheal intubation and who died were consistent with other reports in the literature. This begs the question then if the IRCU had a positive impact or not. A conclusion was that “management of these patients in an IRCU could reduce the need for intubation and consequently reduce complications and mortality.” There is no data to support this. If true, this would be novel and reportable.

Page 1 (title), page 2 (abstract title), page 16 (row 6).

We fully agree with this comment. The intermediate respiratory care unit is nothing more than the facility where respiratory therapies were applied in the absence of resources in the intensive care unit. We have made the suggested change to the specific contribution of the IRCU in our results. We hope that after several corrections it Will more clearly explained that it was the médium where the activity took place and that it did not contribute by itself to our results. Title: Outcomes of an Intermediate Respiratory Care Unit in the COVID-19 pandemic.

Abstract Title: Outcomes of an Intermediate Respiratory Care Unit in the COVID-19 pandemic.

Discussion: “The main finding of this study is that in 37.1% of patients with severe respiratory failure due to COVID-19 managed with non-invasive respiratory support in the IRCU ETI was needed, observing a significantly lower complication and death rates at 28 days in patients were intubation could be avoided.”

Conclusions: “The management of these patients with non-invasive respiratory support in an Intermediate Respiratory Care Units could reduce the need for intubation and consequently reduce complications and mortality”.

5 The manuscript would be improved if the covid patients admitted to the IRCU were compared with the outcome of covid patients admitted at the same time to the ICU. Presumably, but not necessarily the ICU patients had more co-morbidities and greater severity of disease. However, if the capacity of the ICU was overwhelmed, the severity of patients admitted to the IRCU may have been closer to the severity of the concomitant ICU.

Unfortunately, we do not have the data of the patients admitted to the ICU, since the study was not designed with the objective of comparing these two groups. Given the collapse of ICUs in the Madrid area during the first wave of the COVID 19 pandemic, very serious patients as defined by a PaO2/FiO2 ratio below 100 mmHg and SAPS II score above 30 points were managed in the IRCU. As such, all patients included in our cohort would meet ICU admission criteria. 

6 The authors do not explain how missing data is accounted for in the analysis. Were these patients excluded? Was data assumed to be missing at random. There is missing data for 13 patients for IL6, which is ~ 19% of the cohort. The authors then found a significant association with IL6 levels. Additional missing data includes procalcitonin (11), another significant association, and Pa02/FiO2 (15 patients), which wasn’t significant

Page 18 (3rd paragraph, row 17).

Missing data was assumed to be missing at random. Despite having made a prospective design, there were some losses mainly attributable to changing protocols based on continuously updated scientific evidence and to healthcare collapse. In view of this possibility, a loss to follow-up of 20% of the cases was estimated to maintain statistical power.

The findings concerning IL-6, procalcitonin and PaO2/FiO2 ratio must be interpreted with caution because of lost data, which was assumed to be missing at random.

7 Why were 22 patients not candidates for endotracheal intubation? DNI orders? Further explanation of their exclusion could be helpful

Page 6, 2nd paragraph, row 13.

These patients were not considered to be candidates for endotracheal intubation according to the ethics committee.

The document "Ethical considerations of the Department of Clinical Bioethics of the Infanta Elena University Hospital about the management of patients who may require assistance in intensive care units", created and approved in March 2020 for this pandemic situation, defines the criteria that have been taken into account to assess which patients are not admitted to the ICU: A Barthel score below 60 points and a 10-year survival estimated by the Charlson index below 25%.

A reference to the ethics committee approval has been added.

---

## [Decision Letter · Decision Letter 1]

2 Dec 2020

Outcomes of an Intermediate Respiratory Care Unit in the COVID-19 pandemic

PONE-D-20-22016R1

Dear Dr. Carrillo Hernández-Rubio,

We’re pleased to inform you that your manuscript has been judged scientifically suitable for publication and will be formally accepted for publication once it meets all outstanding technical requirements.

Kind regards,

Manjula Karpurapu

Academic Editor

PLOS ONE

Additional Editor Comments (optional):

Reviewers' comments:

Reviewer's Responses to Questions

**Comments to the Author**

1. If the authors have adequately addressed your comments raised in a previous round of review and you feel that this manuscript is now acceptable for publication, you may indicate that here to bypass the “Comments to the Author” section, enter your conflict of interest statement in the “Confidential to Editor” section, and submit your "Accept" recommendation.

Reviewer #1: All comments have been addressed

2. Is the manuscript technically sound, and do the data support the conclusions?

Reviewer #1: Yes

3. Has the statistical analysis been performed appropriately and rigorously? 

Reviewer #1: Yes

4. Have the authors made all data underlying the findings in their manuscript fully available?

Reviewer #1: Yes

5. Is the manuscript presented in an intelligible fashion and written in standard English?

Reviewer #1: Yes

6. Review Comments to the Author

Reviewer #1: There are no additional or remaining concerns that have not been addressed. The investigators have responded as well as possible to the original concerns of the reviiewers.

7. PLOS authors have the option to publish the peer review history of their article (what does this mean?). If published, this will include your full peer review and any attached files.

Reviewer #1: No

---

## [Editor Report · Acceptance letter]

4 Dec 2020

PONE-D-20-22016R1 

Outcomes of an Intermediate Respiratory Care Unit in the COVID-19 pandemic  

Dear Dr. Carrillo Hernandez-Rubio:

I'm pleased to inform you that your manuscript has been deemed suitable for publication in PLOS ONE. Congratulations! Your manuscript is now with our production department. 

Kind regards, 

on behalf of

Dr. Manjula Karpurapu 

Academic Editor

PLOS ONE